# Prediction Models for Intrauterine Growth Restriction Using Artificial Intelligence and Machine Learning: A Systematic Review and Meta-Analysis

**DOI:** 10.3390/healthcare11111617

**Published:** 2023-06-01

**Authors:** Riccardo Rescinito, Matteo Ratti, Anil Babu Payedimarri, Massimiliano Panella

**Affiliations:** Department of Translational Medicine (DiMeT), University of Eastern Piedmont/Piemonte Orientale (UPO), 28100 Novara, Italy; 10033325@studenti.uniupo.it (R.R.); anil.payedimarri@uniupo.it (A.B.P.); massimiliano.panella@uniupo.it (M.P.)

**Keywords:** artificial intelligence, machine learning, intrauterine growth restriction, fetal growth restriction, prediction models, small for gestational age

## Abstract

Background: IntraUterine Growth Restriction (IUGR) is a global public health concern and has major implications for neonatal health. The early diagnosis of this condition is crucial for obtaining positive outcomes for the newborn. In recent years Artificial intelligence (AI) and machine learning (ML) techniques are being used to identify risk factors and provide early prediction of IUGR. We performed a systematic review (SR) and meta-analysis (MA) aimed to evaluate the use and performance of AI/ML models in detecting fetuses at risk of IUGR. Methods: We conducted a systematic review according to the PRISMA checklist. We searched for studies in all the principal medical databases (MEDLINE, EMBASE, CINAHL, Scopus, Web of Science, and Cochrane). To assess the quality of the studies we used the JBI and CASP tools. We performed a meta-analysis of the diagnostic test accuracy, along with the calculation of the pooled principal measures. Results: We included 20 studies reporting the use of AI/ML models for the prediction of IUGR. Out of these, 10 studies were used for the quantitative meta-analysis. The most common input variable to predict IUGR was the fetal heart rate variability (*n* = 8, 40%), followed by the biochemical or biological markers (*n* = 5, 25%), DNA profiling data (*n* = 2, 10%), Doppler indices (*n* = 3, 15%), MRI data (*n* = 1, 5%), and physiological, clinical, or socioeconomic data (*n* = 1, 5%). Overall, we found that AI/ML techniques could be effective in predicting and identifying fetuses at risk for IUGR during pregnancy with the following pooled overall diagnostic performance: sensitivity = 0.84 (95% CI 0.80–0.88), specificity = 0.87 (95% CI 0.83–0.90), positive predictive value = 0.78 (95% CI 0.68–0.86), negative predictive value = 0.91 (95% CI 0.86–0.94) and diagnostic odds ratio = 30.97 (95% CI 19.34–49.59). In detail, the RF-SVM (Random Forest–Support Vector Machine) model (with 97% accuracy) showed the best results in predicting IUGR from FHR parameters derived from CTG. Conclusions: our findings showed that AI/ML could be part of a more accurate and cost-effective screening method for IUGR and be of help in optimizing pregnancy outcomes. However, before the introduction into clinical daily practice, an appropriate algorithmic improvement and refinement is needed, and the importance of quality assessment and uniform diagnostic criteria should be further emphasized.

## 1. Introduction

Intrauterine Growth Restriction (IUGR) is a major public health concern worldwide and has significant implications for neonatal health. The early diagnosis of this condition is crucial for obtaining positive outcomes for the newborn. IUGR is defined as fetuses that did not reach their expected growth potential without major congenital anomalies. They correspond to approximately 70% of small-for-gestational-age (SGA) fetuses, and their growth curve is characterized by a declining trend (abdominal circumference measured by ultrasound <10th percentile) [1]. IUGR is also a significant global burden and a major health issue in developing countries [2]: it is associated with an increased risk of stillbirth, admission to a neonatal intensive care unit, neonatal mortality, cognitive and behavioral disorders in infancy, and chronic noncommunicable diseases in adulthood [2,3]. The incidence of IUGR varies among countries, with a global annual estimate of 30 million newborns affected by this pathology [2,3]. Fifty-two percent of stillbirths are currently associated with IUGR, and 10% of perinatal mortality is reputed to be a consequence of IUGR [4]. IUGR diagnosis is very complex; its suspicion is based on clinical, biochemical, and instrumental criteria [5,6,7]. Ideally, the diagnosis of IUGR consists of the demonstration of growth restriction by ultrasound and the identification of a specific cause [8]. The diagnostic process includes a detailed maternal and family history regarding risk factors for IUGR, maternal physical examination, cardiotocography (CTG), and Doppler ultrasonography to estimate the gestational age (GA) [9]. By comparing these estimates with normal growth curves in the population, it is possible to determine whether the fetus is maintaining its growth potential throughout the pregnancy. This procedure is characterized by an error of 7–10% [9] and low accuracy is reported in detecting decreased birth weight [10] or neonatal morbidity [5] in the first [11], second [12], or third trimester [13] of pregnancy. Usually, when only clinical or ultrasound techniques are used, most SGA newborns are identified after birth using population-based methods [14] or individual curves [10]. Recently, biomarkers such as placental growth factor (PlGF), soluble fms-like tyrosine kynase 1 (s-Flt-1), and alpha-fetoprotein [15] have been demonstrated to play a role in the IUGR development. However, the performance of these angiogenic factors as predictors is limited (positive likelihood ratio, LR +, of 1.3 for PlGF and 1.4 for s-Flt-1) [16]. Similarly, placental proteins are not sufficiently robust biomarkers for IUGR (e.g., LR + of 3.7 for placental protein-13 in the first trimester) [17]. Moreover, the clinical models based on maternal background and Doppler ultrasound at first or second trimester showed moderate performances [18,19]: for instance, Crovetto et al. [20] reported detection rates of 59% and 67% about FGR with a regression model that considered maternal characteristics, uterine artery Doppler, placental growth factor, and soluble fms-like tyrosine kinase-1. Therefore, the International Society of Ultrasound in Obstetrics and Gynecology (ISUOG) guidelines provide only weak recommendations for the diagnosis and treatment of IUGR [8,9,21].

In this context, artificial intelligence (AI), and/or machine learning (ML), deep learning (DL), ensemble learning (EL), and semi-supervised learning (SSL) have emerged as potential options and solutions for many applications (e.g., image recognition, data mining, natural language processing, and for the diagnostic process [22,23,24]) to improve pregnancy outcomes [25,26], including the diagnosis of IUGR.

Traditionally, the AI/ML techniques are divided mainly into unsupervised and supervised learning, with the latter mostly employed in IUGR diagnosis [27]. Among the most widely used approaches in the literature we found the support vector machines (SVM), neural networks (NN), and Random Forest (RF) [25,28]. However, the interpretability of the resulting predictive models is very different among these algorithms: in fact, for many of those (e.g., SVM or NN), it is not possible to know how the model utilizes the patient characteristics to produce the final diagnosis. This raises two major concerns: the first one is ethical, concerning patients that may not accept an automatic mechanism for his/her diagnosis; the other one is legal, concerning physicians who may feel not responsible for any error that may occur [29].

The application of AI/ML/DL/EL/SSL methods has already shown remarkable performance in other fields of women health care [30,31,32], using different types of data such as clinical data, computed tomography (CT), cardiotocography (CTG), electromyography (EMG), genomic, metabolomic, biophysical, and biochemical data [33,34,35,36].

Therefore, we decided to perform a systematic review and meta-analysis in order to:describe the most widely used AI/ML algorithms applied in IUGR diagnosisevaluate the performance of AI/ML models in terms of accuracydetermine whether there is a combination of these methods that has been shown to provide better accuracy in early diagnosis.

The article is organized as follows: In Section 2 we present the methodology of the study. In Section 3 we report the results, both in qualitative and quantitative terms (meta-analysis). In Section 4 we discuss our study findings, and in Section 5 we draw the conclusions of the study.

## 2. Materials and Methods

Three reviewers (R.R., M.R., and A.B.P.) performed the systematic review according to the PRISMA (Preferred Reporting Items for Systematic Reviews and Meta-Analyses) checklist [37]. In this section we summarize the search strategy, the inclusion/exclusion criteria, the selection criteria, the quality assessment of the included materials, the data extraction and, finally, the statistical plan.

### 2.1. Search Strategy

We conducted a literature search in the following electronic databases: MEDLINE (PubMed), Excerpta Medical Database (EMBASE), Scopus, Cumulative Index to Nursing and Allied Health Literature (CINAHL), and Web of Science and Cochrane Database of Systematic Reviews (CDSR). The search was limited to English-language papers. The MeSH (Medical Subject Headings) terms used were “Machine learning” AND “IUGR” OR “Fetal growth retardation” OR “Fetal growth restriction”. Artificial Intelligence AND “IUGR” OR “Fetal growth retardation” OR “Fetal growth restriction”. We also used a snowball search strategy, that is, the reference lists of selected and published original research articles were manually searched to identify other possible eligible studies. The following search strings were created using the search terms or keywords in the databases to locate the records.

MEDLINE (PubMed): “Fetal Growth Retardation [Mesh]) Machine Learning [Mesh]”, “Fetal growth restriction machine learning”, “Machine learning IUGR”, “(Artificial intelligence) AND (IUGR)”, “(Artificial Intelligence) AND (Fetal growth restriction)”, “(Artificial Intelligence) AND (Fetal growth retardation)”EMBASE: “‘artificial intelligence’/exp AND ‘intrauterine growth retardation’/exp EMTREE”, “‘machine learning’/exp AND ‘fetal growth retardation’/exp” AND “‘machine learning’/exp AND ‘fetal growth restriction’/exp”Scopus: “Fetal Growth Retardation Machine Learning”, “Fetal growth restriction machine learning”, “Machine learning IUGR”, “(Artificial intelligence) AND (IUGR)”, “(Artificial Intelligence) AND (Fetal growth restriction)” and “(Artificial Intelligence) AND (Fetal growth retardation)”CINAHL: “Fetal Growth Retardation Machine Learning”, “Fetal growth restriction Machine Learning”, “Machine learning IUGR”, “(Artificial intelligence) AND (IUGR)”, “(Artificial Intelligence) AND (Fetal growth restriction)”, “(Artificial Intelligence) AND (Fetal growth retardation)”Web of Science: “Machine learning IUGR”, “Fetal growth retardation machine learning”, “Fetal growth retardation machine learning”, “(artificial intelligence) and IUGR”, “(artificial intelligence) and (fetal growth restriction)”, “(artificial intelligence) and (fetal growth retardation)”Cochrane Database of Systematic Reviews: “Fetal Growth Retardation/restriction/IUGR AND Machine Learning”, “Fetal Growth Retardation/restriction/ IUGR AND Artificial intelligence”

### 2.2. Inclusion and Exclusion Criteria

We included all studies involving women during fertile age and studies that investigated the use of AI and/or ML in pregnant women for early diagnosis and predictive indicators of IUGR outcome. We included only English-language articles (published as of November 2022). We considered all types of studies (observational studies, clinical trials, systematic reviews, and meta-analyses). The studies dealing with animals, cost-effectiveness studies, articles published as abstracts, letters to the editor, conference proceedings, and dissertations were excluded.

### 2.3. Selection Criteria

Based on the titles, two independent reviewers (M.R. and R.R.) screened the articles to identify relevant studies. One reviewer (R.R.) assessed all article abstracts for eligibility after the screening of titles was completed. Abstracts that met the inclusion criteria were included in the review. Studies for which one reviewer (R.R.) was unsure whether they should have been included were discussed by two further reviewers (M.R., R.R.). Disagreements between the reviewers were resolved by the inclusion of a third reviewer (A.B.P.) and discussion. The full text of the included articles were reviewed for the AI/ML model, method used, pregnancy time, and performance measures (accuracy). The Covidence tool was used for the article selection process [38].

### 2.4. Quality Assessments of Studies

The quality of all included studies was assessed by two independent reviewers (M.R., A.B.P.) using the JBI (Joanna Briggs Institute) critical appraisal tool and the CASP (Critical Appraisal Skills Programme) tool [39,40]. The JBI Critical Appraisal Tool is commonly used to assess the methodological quality of studies. It consists of ten items that address the internal validity and risk of bias of studies, specifically confounding, selection and information bias, and the importance of clear reporting. The CASP tool, instead, is used to appraise the Clinical Prediction Rules of the studies. Disagreements between the two reviewers were resolved by bringing in a third reviewer (M.P.) and through discussions. A high risk of bias was identified when positive responses were ≤49%; moderate risk of bias was assumed when the measure was between 50% and 69%; low risk of bias was identified when positive responses were more than 70%.

### 2.5. Data Extraction

A data extraction sheet was developed to collect data of interest. Four reviewers (R.R., M.P., M.R., and A.B.P) read all included articles. Two authors (A.B.P. and R.R.) extracted the data from the articles, and later they were double-checked by two other authors (M.R., M.P.). The following information was extracted from each study: title, year, setting, study type, sample size, pregnancy time, methods, AL and/or ML model, outcomes, sensitivity, and specificity measures.

### 2.6. Statistical Analysis

For the meta-analysis, R version 4.1.0 (v. 2021-05-18) [41] and R Studio (v.2021.09.2) [42] with the meta package (v.4.19-2) were used to calculate the overall pooled estimates of sensitivity, specificity, diagnostic odds ratio (DOR) [43], and to generate the relative forest plots. We considered a univariate approach (with logit transformations) along with a random effect model. The confidence intervals were calculated with the Clopper–Pearson method. The I^2^ test was used to assess the statistical heterogeneity between the included studies, with the following interpretation: An I^2^ value greater than 75% indicates large heterogeneity between studies. A value of 50 to 75% was interpreted as high heterogeneity, 25 to 50% moderate, and 0 to 25% low heterogeneity. This values explain the proportion of the aggregate pooled effect that is not explainable by chance alone [44]. We will recommended caution in interpreting aggregate measure results when accompanied with significant heterogeneity (more than 50%). The analysis of included studies was divided into two subgroups and random effects analyses were performed to assess the performance of ML methods in predicting IUGR. The abilities of the various ML algorithms to predict IUGR were discussed for each subgroup.

## 3. Results

This section describes the results of the extracted studies and is organized as follows: first we report the entries retrieved, then we describe the general characteristics of the studies and the qualitative synthesis of the performance of the different AI/ML models. Finally, we provide the results of the quantitative meta analysis.

### 3.1. Study Selection

We found 381 entries in Medline (PubMed), Scopus, Cochrane Library, CINAHL, Web of Science, and EMBASE. We did not find any systematic review or meta-analyses in the Cochrane Library. After removing 226 duplicate entries, 155 articles remained for screening. By reading the title and abstract, 118 entries were excluded because they did not meet our inclusion criteria. Then, we processed 37 full-text articles for eligibility. After reading the full text, 17 articles were excluded. Therefore, 20 studies were included in our systematic review. The entire article selection process is shown in Figure 1. All studies met JBI and CASP criteria for critical appraisal. A summary of the quality assessment results is provided in the Appendix A. In summary, when the quality of studies was assessed with the JBI instrument, 17 studies had a low risk of bias with a high percentage of positive responses to the JBI instrument questions, one study was assessed as moderate risk of bias and two studies had a high risk of bias (Appendix A). The CASP instrument identified 18 studies that had a low risk of bias with a high percentage of positive responses to the CASP instrument questions, whereas two studies had a moderate risk of bias (Appendix A).

### 3.2. Study Characteristics

Table 1 describes the characteristics of the 20 included studies. Most studies (*n* = 18) used data from a single country. Twelve studies used data from Europe (Italy and Slovenia), two studies utilized data from the United States, three studies used data from China, and the last study used data from the United States and South Africa. Considering the study design, 16 studies were retrospective case-control studies; 3 were prospective case-control study and the last was a retrospective, nested case-control study for a cumulative total of 11,800 subjects. In detail, one study had a sample size of 6004 subjects, another one included 2199 subjects. Eight studies had more than 100 subjects, with two studies having more than 500 subjects. Finally, ten studies had fewer than 100 subjects. Three studies were conducted in the second trimester, fifteen studies predicting IUGR were conducted in the third trimester of pregnancy and two studies did not report pregnancy time.

Among the 20 studies, the most important features or variables analyzed to predict IUGR using AI/ML were based on FHR parameters from cardiotocography (*n* = 8, 40%) followed by biochemical or biological characteristics (*n* = 5, 25%), Doppler indices parameters (*n* = 3, 15%), DNA profiling data (*n* = 2, 10%), MRI magnetic resonance imaging data (*n* = 1, 5%), and physiological, clinical or socioeconomic data (*n* = 1, 5%). Regarding the AI/ML models used in the studies to predict IUGR: seven studies used the Support Vector Machines (SVM) algorithm [45,46,47,48], with one study using a combination of the Radial Basis Function (RBF) and SVM algorithms [47], one study used SVM and MLP [49]. Four studies adapted logistic regression (LR) model [27,50,51,52]. Of these, three studies used LR combined with Random Forest (RF) [51], Stochastic Gradient Decent & RF [27], and SVM models [52]. Two studies used multiple models [25,53]. One study used a combination of Auto-Contractive Map (ACM) & Activation and Competition System (ACS) [54]. The remaining studies used a single model including RUS Boost, Bayesian network (BN), Artificial neural networks (ANNS), and Lempel–Ziv (LZ) complexity, CVR [55,56,57,58,59]. Regarding the outcomes, as shown in Table 1, nine studies focused on finding the best classification model ML to detect IUGR. Four studies focused on realizing an automatic system for detecting IUGR, three studies focused on evaluating ML models for predicting IUGR and BW, two studies focused on assessing ML models to predict pre-Eclampsia and IUGR [53,59], one study focused on evaluating ML algorithms to identify latent risk clinical attributes [25] and the last study focused on developing an ML screener for late IUGR. Among the included studies, accuracy (measure) in predicting IUGR ranged from 78% (using the SVM) to 97% (using the RF).

**Table 1 healthcare-11-01617-t001:** Characteristics of included studies: articles published using AI and/or ML to detect IUGR during various pregnancy timings.

Author	Study Type	Setting	Sample (n)	Preg Time (weeks)	Methods	AI/ML Model	Outcomes	Measures (Accuracy)
Guo, Z. [50]	Obs. retr. case-control	China	2199	12–28	DNA profiling	LR	Using ML to predict FGR and BW	79%
Dahdoud, S. [58]	Obs. retr. case-control	USA	80	18–39	MRI	RUSBoost	Using ML to predict FGR and BW	86%
Lunghi, F. [45]	Obs. retr. case-control	Italy	909	30–35	FHR by CTG	SVM	Realizing an automatic system for identified FGR	84%
Magenes, G. [48]	Obs. retr. case-control	Italy	100	30–35	FHR by CTG	SVM	Realizing an automatic system for identified FGR	78%
Signorini, M. [60]	Obs. retr. case-control	Italy	120	30–35	FHR by CTG	RF (best)	Find the best classification ML model for identifying IUGR	91%
Crockart, I.C. [27]	Obs. prosp. case-control	USA and S. Africa	6004	20–29	FHR by CTG	Stochastic Gradient Descent, LR & RF	Using ML to predict FGR and BW	91%
Bahado–Singh, R. [46]	Obs. retr. case-control	USA	80	Delivery	Biochemical	SVM	Find the best classification ML model for identifying IUGR	80%
Pini, N. [47]	Obs. retr. case-control	Italy	262	36–37	FHR by CTG	RBF-SVM	Build a ML screener for late IUGR	93%
Magenes, G. [51]	Obs. retr. case-control	Italy	122	30–35	FHR by CTG	RF & LR	Find the best classification ML model for identifying IUGR	RF = 85%; LR = 83%
Xu, C. [52]	Obs. retr. nested case-control	China	810	12–27	DNA profiling	SVM & LR	Find the best classification ML model for identifying IUGR	83%
Buscema, M. [54]	Obs. retr. case-control	Italy	46	Delivery	Biochemical	ACM & ACS	Find the best classification ML model for identifying IUGR	87%
Foltran, F. [55]	Obs. prosp. case-control	Italy	46	20–32	Biochemical	BN	Realizing an automatic system for identified FGR	90%
Street, M.E. [56]	Obs. retr. case-control	Italy	48	20–32	Biochemical	ANNS	Find the best classification ML model for identifying IUGR	89%
Ferrario, M. [57]	Obs. retr. case-control	Italy	59	27–34	FHR by CTG	LZ complexity	Realizing an automatic system for identified FGR	91%
Deval, R. [49]	Obs. retr. case-control	India	214	-	Biochemical	SVM, MLP	Using ML models to predict IUGR	SMO = 95.5%; MLP = 8.5%
Gómez–Jemes, L. [53]	Obs. retr. case-control	Slovenia	95	24–38	Doppler indices: UA, sFIt-1, and PIGF values	Multi-models (extra-trees, RF)	Using ML models to predict pre-Eclampsia, IUGR	Extra trees = 78%, RF = 73%
Sufriyana, H. [59]	Obs. prosp. case-control	Slovenia	95	24–37	Doppler indices: UA, sFIt–1, and PIGF values	CVR	Using ML models to predict pre-Eclampsia, IUGR	CVR = 90.6%
Aslam, N. [61]	Obs. retr. case-control	Italy	382	30–37	FHR by CTG	SVM & RF	Using ML models to predict IUGR	RF = 97%
Gürgen, F. [62]	Obs. retr. case-control	Turkey	44	<38	Doppler indices: PI & RI of UA, MCA, DV, and AFI	SVM	Using ML models to predict IUGR	SVM = 81%
Van, S.N. [25]	Obs. prosp. case-control	China	75	-	Physiological, clinical, and socioeconomic	Seven ML algorithms	Identify the latent risk clinical attributes using the ML algorithms.	94.73%

### 3.3. Performance of ML Models for IUGR Prediction: Qualitative Synthesis

#### 3.3.1. Prediction of IUGR from Biochemical and Clinical Parameters

Five studies examined biochemical and clinical factor data to detect IUGR using the BN, ANNS, ACM&ACS, MLP and SVM models [46,49,54,55,56]. Three studies detected TNF-alfa, IL-6, and IGF-2 as placental markers of fetal growth in IUGR [54,55,56] using AI/ML models. Of these three studies, the BN model showed high predictive accuracy with 90%, followed by ANN with 89%, and ACM-ACS with 87% respectively. Deval et al. analyzed maternal features and blood biochemical parameters using SMO and MLP models. The study showed a predictive accuracy of 95.5% with support vector machine (SVM) algorithm and 88.5% with multilayer perceptron (MLP) algorithm [49]. Bahado et al. performed a metabolomic analysis (with 40 IUGR and 40 healthy subjects) using the SVM model and detected altered metabolic pathways (beta-oxidation of very long fatty acids, oxidation of branched-chain fatty acids, phospholipid biosynthesis, lysine depletion, urea cycle, and fatty acid metabolism) in IUGR cases. The model showed a predictive accuracy of 80% [46].

#### 3.3.2. Prediction of IUGR from DNA Profiling

Two studies analysed nucleosome profiling data from cell-free DNA (cfDNA) to predict IUGR [50,52]. Guo et al. predicted pregnancy complications associated with placental changes (pre-eclampsia, gestational diabetes mellitus, fetal growth restriction, and macrosomia) using whole-genome sequencing with low coverage of plasma DNA from 2199 pregnancies (with maternal age, BMI, neonatal weight, and adverse events in previous pregnancies being the most influential characteristics in the study). This study suggested that promoter profiling-based classifiers provide high predictive power for predicting IUGR in early gestational age with 79% accuracy using the LR model. Moreover, cfDNA profiling can be used as a biological marker for predicting IUGR in early gestational age [50]. In another study, Xu et al. successfully developed the best classifier (IUGR prediction method) for assessing the risk of IUGR in early gestational age with 83% accuracy using SVM-LR models [52].

#### 3.3.3. Prediction of IUGR from MRI Data

Dahdoud et al. examined 3D (three-dimensional) MRI data (placental shape and texture features) to predict IUGR and BW (birth weight) using the RUSBoost model. The model identified IUGR pregnancies with 86% accuracy, 77% precision, and 86% recall. Birth weight (BW) estimates were 0.3 ± 13.4% (mean percent error ± standard error) for healthy fetuses and −2.6 ± 15.9% for IUGR [58].

#### 3.3.4. Prediction of IUGR from FHR Parameters by CTG

The use of fetal heart rate (FHR) in antepartum fetal monitoring (AFM) to predict health status is widely used in clinical practice. We identified eight studies that used AI/ML to classify IUGR and healthy fetuses based on FHR parameters [27,45,47,48,51,57,60,61]. Two studies investigated 15 data mining techniques to select the most reliable approach to detecting IUGR fetuses [51,60]. Magenes et al. found that RF and LR models had the best classification accuracy (RF = 85% and LR = 83%) and that both models outperformed the best single parameter in terms of mean Area Under the Receiver Operating Characteristics (AUROC) in the test sets [51]. In another study, Signorini et al. obtained the best performance for the model RF with 91% accuracy [60]. Two studies aimed to implement an automated system for the diagnosis of fetal sufferance in IUGR fetuses using an SVM model applied to reliable indices extracted from FHR recordings [45,48]. Lunghi et al. achieved 84% accuracy [45] while Magenes et al. obtained 78% accuracy in detecting fetal sufferance in IUGR fetuses [48]. Crockart et al. developed a Stochastic Gradient Descent model that accurately and consistently predicted IUGR with 91% accuracy between 34 + 0 and 37 + 6 weeks of gestation. In addition, the model identified the Umbilical Artery Pulsatility Index as the strongest indicator for predicting IUGR, which is considered the gold standard in the literature [27]. Ferrario et al. analyzed FHR signals from fetuses whose gestational age was between 27 and 34 weeks to detect severe IUGR by applying the Lempel–Ziv (LZ) complexity model. The study showed that the model may be a potential solution for detecting severe IUGR (with 91% accuracy). LZ values were higher for severe IUGRs (>0.94, i.e., close to 1, the theoretical value assigned to a random string), whereas the LZ value for healthy fetuses ranged from 0.85 to 0.9 [57]. Pini et al. used a Radial Basis Function Support Vector Machine (RBF-SVM) classification model for quantitative features extracted from FHR signals acquired by CTG in a population of 160 healthy and 102 late IUGR fetuses. This study showed a satisfactory classification performance in the training group (accuracy 0.93, sensitivity 0.93, specificity 0.84) and concluded that the model can describe the relationships between features beyond the traditional linear approaches, thus improving the classification performance. Moreover, the study found that the model has the potential to be proposed as a screening tool for identifying fetuses with late IUGR [47]. Aslam et al. tested several ML models for the features extracted from FHR signals acquired by CTG. Among the tested models SVM model had an outperformed positive predictive value (PPV) and Specificity (SP), whereas RF model showed highest predictive accuracy (97%) [61].

#### 3.3.5. Prediction of IUGR from Doppler Indices

We identified three studies that analysed parameters (including UA, sFIt-1, PIGF values, PI & RI of UA, MCA, DV, and AFI obtained using Doppler ultrasound) Using ML models, in which two studies predicted IUGR and pre-eclampsia [53,59] and one study to predict IUGR fetus at risk [62]. Of these three studies, the CVR model showed high predictive accuracy 90.6% [59] followed by SVM model with 81% [62] and Extra trees (with 78%) and RF model (with 73%) respectively [53].

#### 3.3.6. Detection of Latent Clinical Attributes for Children Born under IUGR Condition

Van et al. study proposed ML as a solution to identify key characteristics based on physiological, clinical, or socioeconomic factors that correlate with previous IUGR status after 10 years. A dataset of 75 children (41 IUGR and 34 non-IUGR subjects) was used for the study. This study showed that the model ML (proposed classification system) had an accuracy of up to 94.73%. The study concluded that the proposed classification system and the indication of relevant attributes (latent factors such as HRV and BP monitoring) may assist medical teams in the clinical monitoring of IUGR children during their childhood development [25].

### 3.4. Meta-Analysis

A total of ten studies were included in the meta-analysis. Different parameters were employed as inputs for AI systems: four studies used biological profiling parameters as AI inputs to predict IUGR [46,50,52,54], six studies used FHR parameters as AI inputs to predict IUGR [27,45,47,48,51,60]. We were unable to extract precise quantitative data from ten studies; thus, they were excluded from the meta-analysis [55,56,57,58].

The results showed that AI/ML models applied to FHR by CTG and biological profiling features for the detection of IUGR had comparable sensitivity and specificity. In particular, the sensitivity was 0.81 [0.73–0.87] for the biological profiling versus 0.86 [0.80–0.90] for the FHR by CTG; the specificity was 0.83 [0.73–0.87] for the biological profiling versus 0.89 [0.85–0.92] for the FHR by CTG. The AI/ML algorithms were able to predict IUGR with a pooled overall sensitivity, specificity, PPV and NPV of 0.84 (95% CI: 0.80–0.88) [Figure 2], 0.87 (95% CI: 0.83–0.90) [Figure 3], 0.78 (95% CI: 0.68–0.86) [Figure 4] and 0.91 (95% CI: 0.86–0.94) [Figure 5] respectively. The pooled diagnostic odds ratio was 30.97 (95% CI: 19.34–49.59) [Figure 6]. We noticed no significant differences between the two groups concerning the pooled sensitivity and specificity. Our results showed no differences between the predictive values (both positive and negative) of the two groups as well. When comparing the aggregate pooledpredictive values, we noticed that the NPV is substantially greater than the PPV, thereby indicating a possible more important role of AI/ML models in detecting an healthy rather than an unhealthy fetus. Finally, the DOR summarizing measure showed also a moderate heterogeneity (26%).

## 4. Discussion

In current practice, the diagnosis of IUGR is a process whose outcomes could be affected by the complexity of the procedure [36,63].

Previous studies [18,19,20] showed moderate performance of clinical models that are based on maternal background and Doppler ultrasound at first or second trimester. Given the results of these studies, the use of AI and machine learning could be an improvement of current models [64]. The quality of ultrasound examinations depends on the operator’s skills and requires repeated measurements during pregnancy. Therefore, the precision of the consequent fetal growth curve could be affected by the moderate reliability of the method [1,8,9].

The results of our meta-analysis showed that AI/ML techniques could help to better predict IUGR than conventional methods and reduce the intrinsic bias in image interpretation, both in the general population and at-risk groups. However, we obtained our findings by grouping several algorithms applied to the different diagnostic techniques and this could be a first limitation of the study. We observed high performances for all the AI/ML models, with an accuracy rate of the diagnostic process ranging from 79% to 97% [47,50,61]. However, it is important to note that the inputs of the AI/ML assisted models were not completely identical among the studies included in the analysis and therefore the interpretation of the aggregate diagnostic performances should be cautious, despite the low/moderate heterogeneity values.

In overall, we observed that AI/ML models better predicted IUGR when applied to CTG. In detail, the RBF-SVM model showed an accuracy of 93%; the LZ-Complexity, Stochastic Gradient Descent, LR, and RF models had an accuracy of 91%. Lastly, the detection of TNF-alfa, IL-6 and IGF-2 as placental markers of fetal growth in IUGR using the BN model had an accuracy of 90%. Therefore, these results suggest that AI/ML could help clinical decisions for birth planning in terms of timing and detection of a more appropriate structure in fetuses at risk of IUGR. Nevertheless, it is difficult to reach a consensus about the best method to predict IUGR because not all the studies included in our review had the same input variables and samples. However, we think that FHR by CTG might lead clinicians to prefer this method in the near future because of its actual widespread use and the simplicity of its application in everyday practice [65].

It is also important to mention that all the studies reviewed were observational and not applied in a clinical setting. To our knowledge, there is no randomized controlled trial (RCT), or pilot study conducted in a medical centre or adaptation of an AI/ML-based application (to predict IUGR) in a hospital. In this regard, to obtain a higher level of evidence, we think that future prospective studies, in particular RCTs with large samples, are needed to confirm the performances in IUGR prediction. These studies should be based on a rigorous comparison between AI/ML models for IUGR prediction (interventions) and usual care (i.e., conventional diagnostics as a control) and necessarily use standardized outcomes to better compare the different AI/ML models for IUGR early diagnosis and to effectively generalize the eventual findings [66,67]. In fact, from a methodological perspective, in order to maximize the performance of AI/ML models the selection of appropriate of input parameters/features is needed [68]. Therefore, the actual lack of studies comparing AI/ML and conventional diagnostic could have affected the actual selection of informative, discriminative, and independent features and limiting the performances of algorithms when applied in clinical setting.

Interestingly, we found that the most accurate algorithms (e.g., SVM) provide a model that is not interpretable, therefore acting as a black box. Even though this aspect is not as important as the absolute accuracy of the diagnosis, ethical concerns may arise because the patients must be informed and accept the fact that not only the physician is not making the diagnosis anymore, but also that he/she could not explain how the decision was made. Finally, we noticed that there are no large scale validation studies in the literature, which we believe are necessary before employing these techniques in current practice.

## 5. Conclusions

Our study provided a first overview of the role of AI/ML technology in predicting IUGR and demonstrated its promising potential. Despite a few limitations, we think that our review can be considered a robust and high-quality study for some reasons. First, the most popular scientific databases were searched with a well-developed query. The search was sensitive and accurate. Then, the use of techniques such as searching the biomedical literature database (PubMed, Embase, and Scopus), the world’s leading citation databases (Web of Science), subject-specific databases (CINAHL), and the world’s leading source of journals and databases for systematic reviews in health care (CDSR) suggests that the risk of publication bias in this study is low. Lastly, most of the studies included in our review had a low risk of bias when assessed with the JBI and CASP tools.

We believe that integrating AI with conventional noninvasive methods could provide doctors better diagnostic tools with an optimal balance of sensitivity and specificity. However, we also think that stronger studies are needed to better validate AI/ML models for prediction of IUGR before their introduction into clinical practice, given the high variability of variables included in all the studies and the possibility of lack of consideration of significant confounders (e.g., pre-eclampsia). These new studies will need to be designed not only to evaluate the efficacy, but also to solve the issue of medical liability when applying AI/ML model in clinical practice. This because some application of AI/ML models is closely related to what is called a black box system [69,70], as the doctor views only input and output parameters and cannot see the inner workings of the algorithm. It is a rather controversial system and until all these processes are made transparent and understandable, even if potentially effective, their application will be limited. Consequently, future research will have to solve this issue as well before this technology can be widespread in everyday clinical practice. Moreover, the introduction of routine use of AI/ML based systems in health care facilities could lead to a better process performance and probably, to cost-saving results. However, specific process indicators measuring AI performances in clinical practice have yet to be developed.

In conclusion, our findings showed that AI/ML methods are promising and could be part of a more accurate and cost-effective screening method for IUGR and be of help in optimizing pregnancy outcomes. However, before the introduction into clinical daily practice, an appropriate algorithmic improvement and refinement is needed and the importance of quality assessment and uniform diagnostic criteria should be further emphasized.

## Figures and Tables

**Figure 1 healthcare-11-01617-f001:**
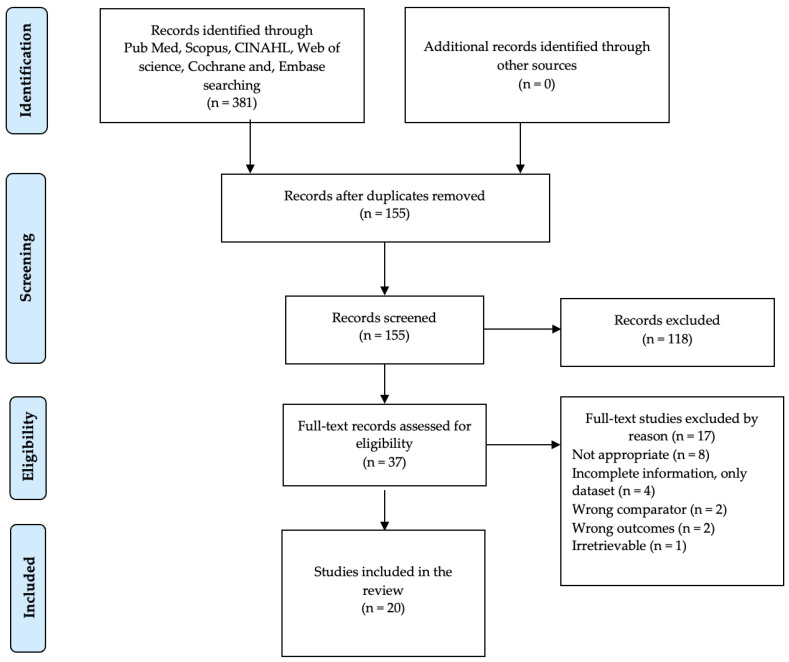
PRISMA Flow Diagram for the selection of articles.

**Figure 2 healthcare-11-01617-f002:**
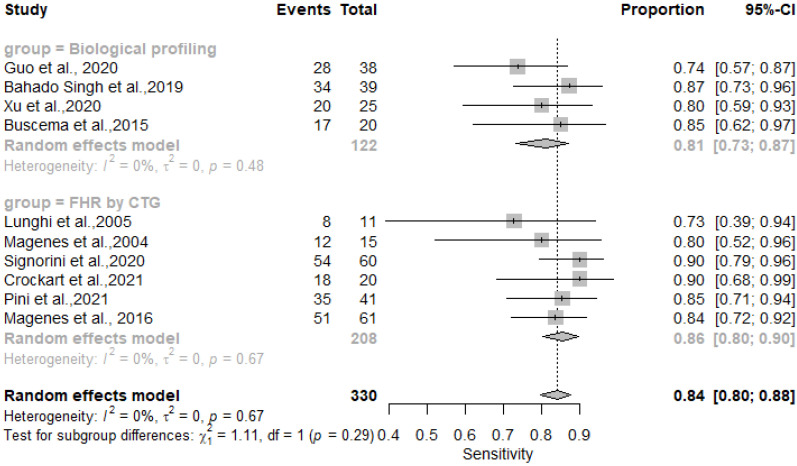
Pooled sensitivity of artificial intelligence and/or machine learning models for IUGR prediction [27,45,46,47,48,50,51,52,54,60].

**Figure 3 healthcare-11-01617-f003:**
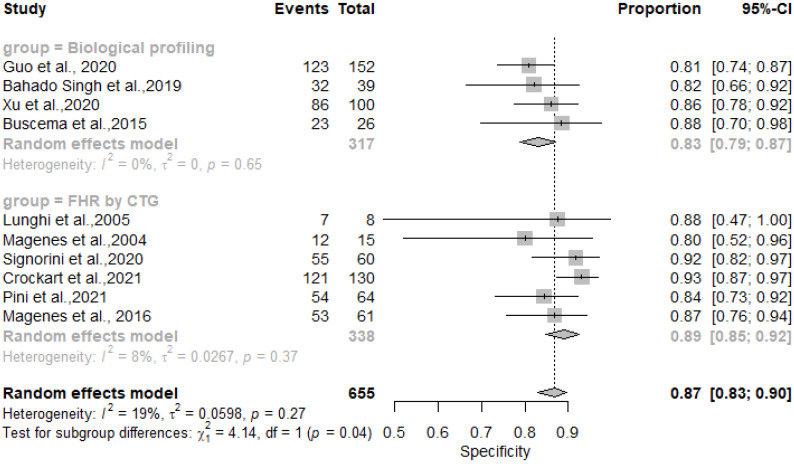
Pooled specificity of artificial intelligence and/or machine learning models for IUGR prediction [27,45,46,47,48,50,51,52,54,60].

**Figure 4 healthcare-11-01617-f004:**
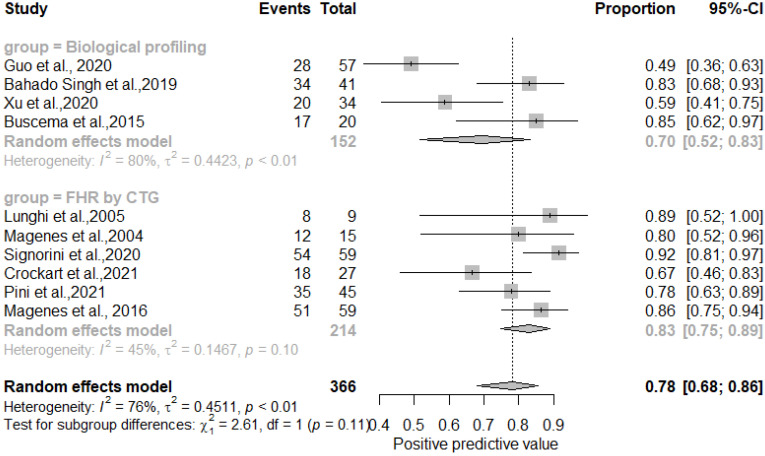
Pooled positive predictive value of artificial intelligence and/or machine learning models for IUGR prediction [27,45,46,47,48,50,51,52,54,60].

**Figure 5 healthcare-11-01617-f005:**
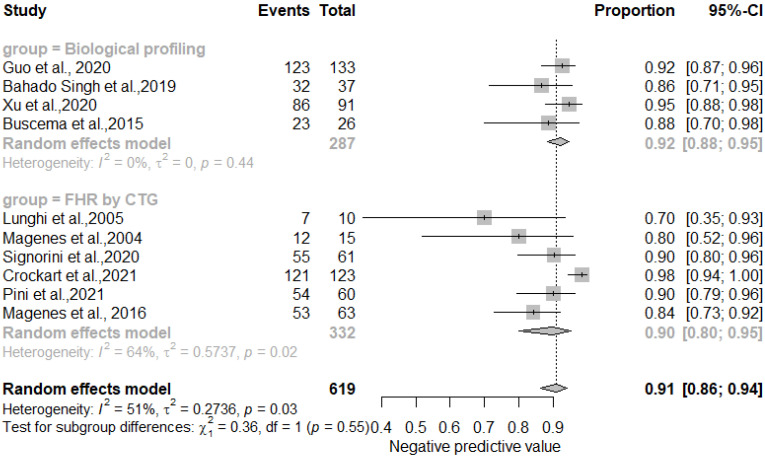
Pooled negative predictive value of artificial intelligence and/or machine learning models for IUGR prediction [27,45,46,47,48,50,51,52,54,60].

**Figure 6 healthcare-11-01617-f006:**
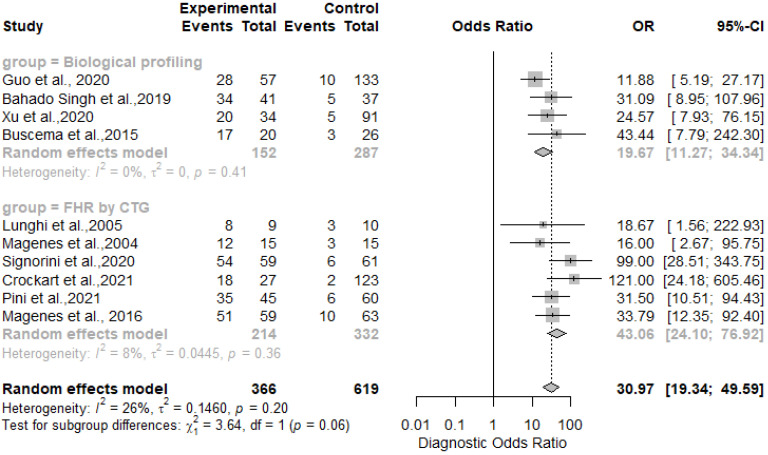
Diagnostic odds ratios of artificial intelligence and/or machine learning models for IUGR prediction [27,45,46,47,48,50,51,52,54,60].

## Data Availability

All data are available upon motivated request.

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
