# Peer review of "Prediction Models for Intrauterine Growth Restriction Using Artificial Intelligence and Machine Learning: A Systematic Review and Meta-Analysis"

_healthcare, 2023, doi:10.3390/healthcare11111617_

Round 1

Reviewer 1 Report

Thank you for the opportunity to review this research paper. Although the topic of this research study is interesting but, I think authors should apply the comments indicated below to increase the quality of research justification, contributions, and findings.

First of all, the similarity report indicates a high level of similarity (49% https://www.turnitin.com)  so I suggest reducing the similarity first of all.

Abstract

A better presentation of the aim of the paper and what is new and original for this paper.  And maybe you can synthesized the abstract.

Introduction

What is the originality of this research?  Paper research gap and originality should be better presented at the end of introduction section.

I would like to suggest the following references:

Akram, U et al. (2021). Impact of digitalization on customers’ well-being in the pandemic period: Challenges and opportunities for the retail industry. International Journal of Environmental Research and Public Health, 18(14), 7533.

Ionescu, C. A., Fülöp, M. T., Topor, D. I., Căpușneanu, S., Breaz, T. O., Stănescu, S. G., & Coman, M. D. (2021). The new era of business digitization through the implementation of 5G technology in Romania. Sustainability13(23), 13401.

Material and methods must be reorganized because the similarity is very high pleas see the attached document.

The results are ok but also here pleas review the similarity.

Discussions are well.

Conclusions: pleas add theoretical, managerial, and practical implications, limitation and further research. Some parts are included but must be extended.

Good Luck!

Author Response

Dear Reviewer,

kindly refer to the attahed document,

regards,

Reviewer 2 Report

Thank you for the opportunity to review this interesting work. The authors have conducted a systematic review and meta-analysis to evaluate the use and performance of AI/ML models in detecting fetuses at risk of IUGR. Their findings suggested that AI/ML methods could be helpful and be a part of a more accurate and cost-effective screening method for IUGR which can further help in optimizing pregnancy outcomes.

Overall, the authors have done a nice job! The systematic review methodology has been done well and described in detail. The authors have assessed risk of bias in the studies which have been presented well. The results from the meta-analysis are clearly described. The limitations and future directions have been listed well.

Minor Comments:

·       When is IUGR typically diagnosed? Are there any clinical recommendations or guidelines? Is it usually in the second or third trimester?

·       Line 105: probably cite the reference for the latest PRISMA guidelines unless the protocol was developed using the previous guidelines.

“Page M J, McKenzie J E, Bossuyt P M, Boutron I, Hoffmann T C, Mulrow C D et al. The PRISMA 2020 statement: an updated guideline for reporting systematic reviews BMJ 2021; 372 :n71 doi:10.1136/bmj.n71”

·       Line 146: were women undergoing fertility treatment excluded? If so, please mention this in the exclusion criteria along with the reason.

Author Response

(The authors gave the same response as above.)

Reviewer 3 Report

Dear colleagues,

I can not accept the article in its current form, mainly because the introduction and discussion are lacking of clinical understanding of IUGR/FGR and the diagnostic methods. This must be improved, maybe you should search counseling by a clinician.

Kind regards!

Must be improved.

Author Response

(The authors gave the same response as above.)

Round 2

Reviewer 1 Report

Good luck!

Author Response

Dear Reviewer,

We thank you for your precious comments and wish you luck as well. 

Kind regards,

Reviewer 3 Report

Dear Sir or Madam,

I have some comments to make:

Line 41: "SGA below the 10th week of gestation". Typically you don't call a fetus SGA in the first trimester.

Line 44 ff: "the diagnostic process includes a detailed maternal and family history regarding risk factors for IUGR, maternal physical examination, cardiotocography (CTG), and Doppler ultrasonography to  determine the exact gestational age (GA) [9]. Using these parameters, fetal weight can be estimated." The process is correct, but fetal weight is just estimated by the sonographic parameters.

Line 377f: "However, we think that FHR by CTG will lead clinicians to prefer this method in the near future because of its actual widespread use and the simplicity of its application in everyday practice [66]." Might, not will.

Kind regards.

Moderate editing of English language required.

Author Response

Please refer to the attached document.

Kind Regards,
